# Fingerprinting Sediment Origin of the Silting Process of Urban Reservoirs

Maria E. A. Ferreira [ID], Diego A. Zanoni, Glauber A. Carvalho, Jamil A. A. Anache [ID], Paulo Tarso S. Oliveira *[ID] and Teodorico Alves Sobrinho [ID]

Faculty of Engineering, Architecture and Urbanism and Geography, Federal University of Mato Grosso do Sul, Campo Grande 79070-900, MS, Brazil
* Correspondence: paulo.t.oliveira@ufms.br; Tel.: +55-67-9-9179-9709

**Abstract:** The silting process of urban reservoirs has been occurring in many regions across the world. However, identifying the main sources of sediment and controlling the silting process in urban reservoirs are still unsolved problems in many regions, mainly in developing countries such as Brazil. In this study, we identify which land use most influences the siltation of reservoirs, and how the different tributary streams contribute to this process in two urban reservoirs located in Campo Grande, Midwestern Brazil. Thus, we applied a sediment source fingerprinting (SSF) approach, associated with land use analysis, and the bathymetric data of reservoirs connected to the stream and drainage network, obtained between the years 2008 and 2018. The reduction in the volume and area of the reservoir during the study period were 45% and 39%, respectively. We found a proportional relationship between the reduction in the reservoir volume and the increase in the impermeable areas of the studied basin. We also noted that the sediments deposited in the reservoir originate from bare soil, banks, and bed in the proportions of 46.9%, 37.1%, and 17.2%, respectively. Our findings show that the use of bathymetric surveys and data on land use and land cover, associated with the source tracing technique, are useful alternatives to identifying sediment mobility in urban basins, especially in those where the drainage network is connected to water courses. We conclude that the factors that most contribute to the silting up of reservoirs are the erosion of banks and beds, sediment remobilization and the connectivity of the drainage network with water courses.

**Keywords:** sediment production; urban catchment; land use and land cover; sediment sources; soil erosion

## 1. Introduction

Urban catchments are subject to rapid changes in land use and occupation, which influence hydrological processes. It is known that soil sealing, increased runoff flows, and changes in vegetated areas are factors that interfere with sediment production in watersheds. In many cases, water bodies are directly affected by urbanization, receiving higher rates of sediment, and undergoing structural changes in beds and channels. The silting up of urban reservoirs is the most visible impact of the urbanization process. Reservoirs, whether natural or built, naturally retain sediments; they can receive 100 times more sediment from uplands when the land cover is urban, compared to forest or pastureland cover [1]. Extreme precipitation events, associated with climate change, are another factor that increase the sedimentation rate in urbanized areas [2]. The impact of urbanization and sediment production on water resources has been addressed over the years by several authors in studies on various topics: on the useful life of reservoirs [3]; on sedimentation analysis [4]; on sediment budget [5,6]; on the impacts on water quality [7], and on source tracing [8].

The evaluation of the silting of reservoirs must consider the load and the annual rate of sediment as the main factors [4]. To reduce the impact, sediment sources must

be found and mitigated [9]. Thus, identifying sediment sources, through tracing, allows planning activities to control erosion in the basin [10]. Source tracing, a technique known as sediment source fingerprinting (SSF), consists of mapping and identifying the origin of sediments that reach water bodies; this is through physical and chemical analyses of the basin's soils and its suspended sediments. In other words, this approach is based on the collection of sediment samples, in order to conduct a comparison between their main properties or composite fingerprints, and to then estimate the relative importance of upstream sources [11]. With the beginning of tracer metal research in the 1970s, the technique proved to be advantageous, compared to the traditional methods of monitoring sediment production [12]. This method underwent improvements and adaptations and has been used worldwide, resulting in publications in more than 150 journals [13]. The SSF technique has already been applied to studies in agricultural basins [14,15], rural and urban basins [16,17], only urban basins [18–20], and in lake-bottom sediment deposits [21]. In urban environments, application difficulties can include changes in land use, which occur in a dynamic and accelerated manner, and diffuse the pollution of metals from anthropogenic sources [22].

We identified the potential sources of sediment in an urban catchment, responsible for the siltation of a reservoir, by using SSF techniques coupled with other environmental dataset analyses (land use and reservoir bathymetric data timeseries). Thus, we characterize the potential sources of sediment responsible for the siltation process in urban reservoirs, considering the changes in land use and occupation, in order to identify possible drivers that might contribute to silting. Data from bathymetries, tracing of sources and a temporal analysis of land use and occupation were associated. At the end, we could reach our main purposes: understand which land use most influence the siltation of reservoirs, also evaluate how the different tributary streams contributed to the process.

## 2. Materials and Methods

### 2.1. Study Area

The study was carried out in two urban catchments in the municipality of Campo Grande, MS, the Bandeira (B) Stream catchment and the Cabaça (C) Stream catchment, which have undergone alterations resulting from the urbanization process. The catchment region has a tropical dry and wet climate (Aw, Köppen), with a hot and wet summer and a dry winter [23]. The average annual rainfall and temperature are 1455.3 mm yr$^{-1}$ and 23.3 °C, respectively [24]. The study area elevation ranges between 529 m and 630 m (a.m.s.l.). The Bandeira (B) stream has a length of 4.6 km, with 0.7 km of canalized bed, and the remaining 3.9 km possessing preserved springs and vegetation on the banks. The Cabaça (C) stream has a length of 2.4 km, 0.8 km of which is completely canalized (working as a stormwater gallery), and its banks are practically devoid of vegetation. The catchments of streams are undergoing a different urbanization process. The Cabaça (C) stream catchment is densely occupied, while the Bandeira (B) stream catchment is in the process of urban densification (Figure 1). Both catchments started their urbanization processes in the 1970s [25].

In the catchments, there are two dammed reservoirs that were built in the 1970s for landscape purposes, without any hydraulic structure that would allow sediment flushing. The main reservoir (MR) (original area of approximately 110,000 m$^2$) is supplied by the Bandeira (B) and Cabaça (C) streams, built at the confluence of the streams, and is in the process of silting. The secondary reservoir (SR) (original area of approximately 20,000 m$^2$) is upstream of the reservoir, it is supplied only by the Bandeira (B) stream, and is currently silted up. The main reservoir, which is the one studied in this article, is located 2.36 km before the outlet of the Bandeira (B) hydrographic basin.

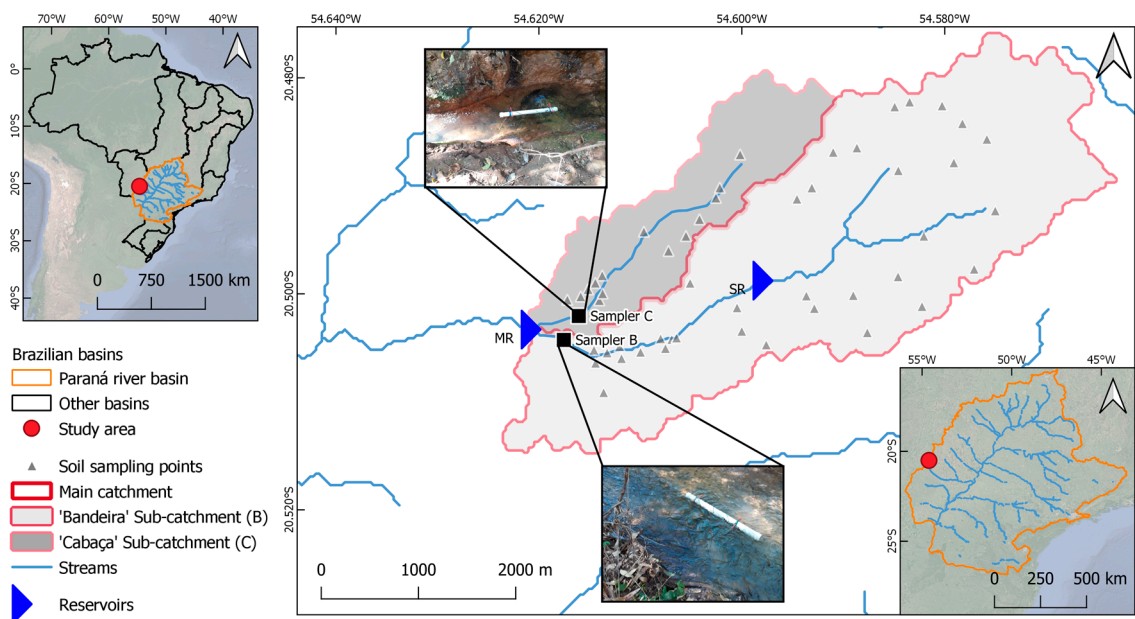

**Figure 1.** Study area, watercourses present in the basin and sampling site of suspended sediments and soil.

In this study, we considered only the drainage area that contributes to the reservoir under study; this area has a length of 12.65 km$^2$, of which 10.62 km$^2$ is drained by the Bandeira (B) stream and its tributaries, and 2.03 km$^2$ is drained by the Cabaça (C) stream and its tributaries. To identify the main sediment sources in the evaluated basins that have contributed and continue to contribute to the siltation of the urban reservoir, we used data from a series of bathymetric surveys, a source tracing technique (SSF), and a temporal analysis of land use and occupation.

*2.2. Bathymetric Data*

To assess the area and volume of the lake (MR), we monitored the bathymetry in the main reservoir (MR) from 2008 to 2018. In total, eight bathymetric surveys were performed. The first bathymetry took place in 2008 using a depth gauge. The other bathymetries that took place in November 2011, February and November 2013, October 2014, March 2016, March 2017, and November 2018 were performed using an acoustic profiler, (SonTek RiverSurveyor M9, SonTek, San Diego, CA, USA) and a boat as a platform to navigate and obtain the data. The bathymetric survey performed in 2008 was considered as a reference due to methodological standardization; the sample points that were randomly obtained considered a density of approximately 0.05 points/m$^2$, and depth measurements with a 1% of accuracy. The bathymetric data was acquired from the acoustic profiler using RiverSurveryor LIVE (RSL) Software (SonTek).

The bathymetric data processing was performed using Surfer® (Golden Software, LLC, Golden, CO, USA). The collected data were modeled by the Nearest Neighbor geostatistical method to obtain the digital elevation model (DEM). From the DEM, generated for each bathymetry performed, we obtained the volume and area of the reservoir. The MR water level is constant as it is controlled by a bell-mouth spillway, which is the only reservoir outlet. Thus, the volume calculation is easily reached, as the reference water level is constant. The area was determined for each annual DEM by considering the sum of pixels that have values below water level. Afterwards, we estimated the useful lifetime of the main reservoir with the temporal analysis and linear regression of the volume and area data.

### 2.3. Characterization of Sediment-Producing Sources

The choice of sediment-producing sources, called primary sources, was based on the hydraulic characteristics and occupations of the basin. In the study area, there are occurrences of impermeable soil by constructions, areas with bare soil and the presence of unpaved roads. The margins of the water courses have fragmented vegetation and urban interference, such as canalization, road crossings, and the contribution of the drainage network.

The three potential sediment-producing sources were named as follows: (i) Bare soil—areas of undergrowth and bare soil in the basin, including unpaved streets; (ii) Bank—comprising vegetated and non-vegetated areas on the riverbanks; and (iii) Bed—consisting of the bed of water courses and the sediment banks formed in the channel.

The collection of samples from potential sources took place between August and October 2019. Moreover, 500 g of surface soil samples were collected. We used non-metallic materials during collection and analysis to avoid possible sample contamination. We collected 51 simples from potential sources, being 23 in bare soil, 20 in banks, and 8 in beds. As the inter-rill soil erosion process occurs in the soil surface, we collected samples from 10 cm deep.

To identify the contribution of potential sediment sources in the siltation of the main reservoir, we collected suspended material in each tributary with the installation of sediment samplers before the confluence of streams in the reservoir. Having identified the samplers, we obtained time-integrated suspended sediment samples with representative particle size distribution for small water bodies [26]. The collection of suspended material, using the sampler, took place in the rainy season, between August 2019 and March 2020.

The suspended sediments were collected with the aid of an integrated time sediment sampler. The sampler has a cylindrical shape, with inlet and outlet holes measuring approximately 10 mm, allowing the sediments, carried by the water, to be deposited in the path between the inlet and the outlet of the sampler. The sampler was installed just below the water surface, preventing the entry of bottom sediments, and collecting only suspended sediments.

Among the possible methods of metal analysis, in this study, we chose the atomic absorption spectrometry (AAS) method, which is the method used in 63% of sediment source tracing studies [27]. Among the tracers, we selected metals of geogenic origin, due to the diversity of probable elements [28], and for the better discrimination of sources in urban areas when compared to metals of anthropogenic origin [29].

The particle size distribution and metal concentrations were obtained in laboratory tests for soil and sediment samples. The strainers used for particle size characterization had holes ranging between 2 and 0.063 mm [30]. We separated the material <0.063 mm for chemical analysis, in order to determine the concentration of metals by atomic absorption spectrometry, using the acid digestion method ($HNO_3$—$H_2O_2$). The metal analysis included the following elements: Lead (Pb), Chromium (Cr), Copper (Cu), Manganese (Mn), Nickel (Ni), Magnesium (Mg), Zinc (Zn), Iron (Fe), Aluminum (Al) and Cadmium (Cd).

The assessment of the conservative or non-conservative property of the tracers was carried out in this study from the concentration range of the potential sources; if the tracer concentration in the sediment sample was outside the concentration range verified in the potential sources, the tracer was excluded for the other tests [31].

Having obtained the concentrations of metals from the sediment-producing sources and the suspended sediments, transported by watercourses, we applied the source tracing method: SSF. This method aims to identify the relative contribution of each sediment source using the concentration of metals as tracers. The method is performed in three stages; the first and second stages involve the selection statistics of tracers, in order to define which of the analyzed metals have the potential to be a tracer [32]. First, we performed the Kruskal–Wallis test, considering the null hypothesis ($p < 0.05$) that the sources belonged to the same population, and then we performed a discriminant function analysis (DFA) to

assess which combination of the metals selected in the previous step had the potential to maximize source discrimination [33].

After considering the definition of the set of discriminant variables, we proceeded to the step of determining the relative contribution of each source from the multivariate model, described by Equation (1).

$$y_i = \sum_{s=1}^{n} a_{is} P_s \qquad (s = 1, 2, \ldots, n) \text{ and } (i = 1, 2, \ldots, m) \tag{1}$$

where $y_i$ is the concentration of metal tracer found in suspended sediment, $a_{is}$ is the sum of the tracer concentrations in the sources, and $P_s$ is the respective proportions of tracer concentrations in the source.

The model finds the solution from an objective function Equation (2), where the function is minimized with restriction values so that it must be greater than or equal to 0 and less than or equal to 1; the sum must always be equal to 1 [34,35]. This step was performed in a spreadsheet using an optimization algorithm. The model presents acceptable solutions when the mean absolute error (MAE) is less than 20%.

$$\sum_{i=1}^{m} \left\{ \left( C_i - \left( \sum_{s=1}^{n} a_{is} P_s \right) \right) / C_i \right\}^2 \tag{2}$$

where $C_i$ is the concentration of tracer i in the suspended sediment, $a_{is}$ is the concentration of tracer $i$ in sources, and $P_s$ is the proportion of tracer concentration in the source.

### 2.4. Land Use and Land Cover

Changes in the catchments were evaluated by a temporal analysis of land use and land cover (LULC) from 2011 to 2019, using RapidEye (images courtesy of Geo Catálogo MMA, Brazil) and Sentinel-2 (images courtesy of the U.S. Geological Survey) imagery, with 5 m and 10 m of spatial resolution, respectively. The visible spectral bands (red, blue and green) were used to classify the LULC throughout the years. The LULC was carried out via the supervised classification of images, which consists of segmenting and classifying the objects and evaluating the accuracy of the classification afterwards. We used the Semi-Automatic Classification Plugin from QGIS Python Plugins Repository to classify all used satellite imagery [36]. We consider four classes of LULC: arboreal vegetation (trees and shrubs), undergrowth (grass), bare soil (soil without vegetative cover), and urbanized (impermeable areas). LULC maps were generated for each year, individually, and after classification, the respective classes were quantified.

All the data collected—historical series of bathymetries, soil samples, suspended sediments, SSF technique for tracing sediment sources, and the temporal analysis of LULC—were analyzed together to support additional information on the process of siltation in the reservoir. The joint analysis of the data allowed the characterization of the temporal evolution of the silting process and the identification of which factors influenced the production of sediment in the study area.

## 3. Results

### 3.1. Bathymetric Surveys

After analyzing the data from the bathymetric surveys, carried out between 2008 and 2018, we found a reduction in the volume and area of the MR (Figure 2). The relationships between volume time and area time were obtained by linear regression. In the first survey, the MR had a capacity of 199,225 m$^3$ and a water surface area of 96,354 m$^2$. Over the monitored period, both the volume and area were gradually reduced, resulting in a volume of 110,167 m$^3$ and an area of 58,913 m$^2$ in 2018. In 2016, we observed an increase in the area and volume in the bathymetry. This may be due to bathymetric monitoring that was carried out during the rainy season in the basin, resulting in a full and overflowing reservoir. The depth of the lake was also reduced (Figure 2). In 2008, the maximum depth found was

4.89 m and the average was 2.06 m. In 2018, the maximum depth decreased to 4.32 m and the average decreased to 1.87 m. From the first to the last bathymetric survey, the total loss of volume and area was 45% and 39%, respectively. During the silting process of the reservoir, there was an accumulation of sand granulometry sediment in the flow of streams into the lake. In Figure 2, it can be observed that the sedimentation of this material formed two sandbanks in the MR, and that between 2017 and 2018, the sandbanks had a significant increase in area, mainly in the Bandeira (B) stream outflow.

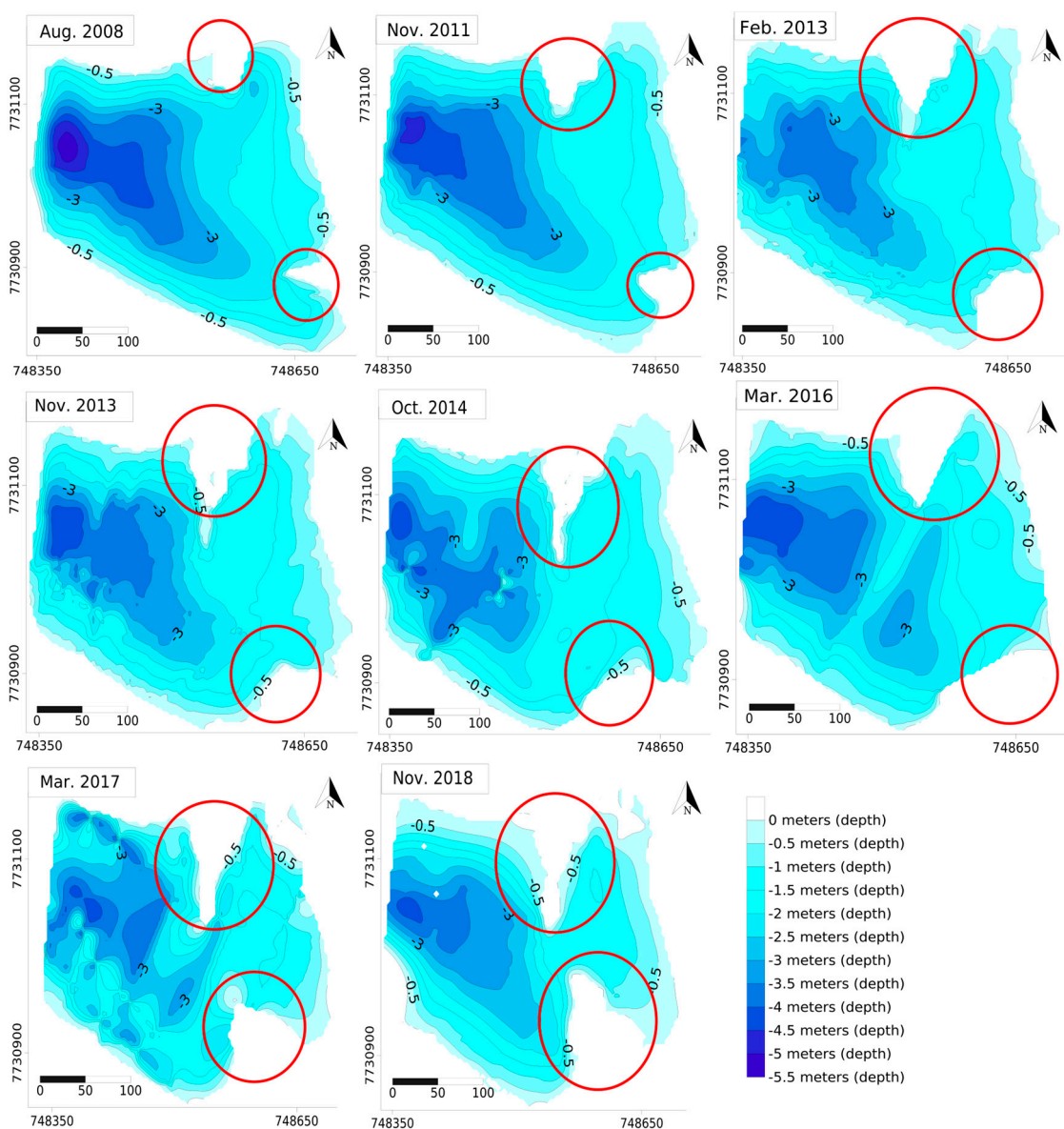

**Figure 2.** Estimates for the loss of volume and area in the main reservoir, based on bathymetric surveys carried out between 2008 and 2018.

### 3.2. Characterization of Sediment-Producing Sources

Fifty-one soil samples were collected from the study area and considered as primary sources. From this total, 23 samples were from bare soil, 20 were from margin samples and 8 were from bed samples. All the primary sources that were analyzed exposed the soil, bank and bed, and showed the presence of metals (Table 1). The samples showed high concentrations of the metals Fe and Al, and Cd was the metal with the lowest concentration sampled.

The particle size distribution indicated that samples from bare soil areas contain an average of 4% of coarse sand, 65% of medium sand, 29% of fine sand and 2% of silt. The bank samples presented 2% of coarse sand, 64% of medium sand, 32% of fine sand and 2% of silt. Bed samples contained 43% of coarse sand, 51% of medium sand, 5% of fine sand and 1% of silt.

Among the proposed tracers, Zn was disregarded for the other tests, as it presented concentrations in the sediment samples above the concentration range of potential sources, indicating that it is not a conservative tracer (Table 1).

The application of the Kruskal–Wallis test in the samples from the primary sources made it possible to select the metals that were capable of being tracers for the sediment-producing sources. Metals that showed discriminating ability were those with a p-value less than or equal to a 5% significance. In this test, Pb, Cu, Zn and Cd were discarded for being unable to distinguish the sources (Table 1).

**Table 1.** Mean metal concentrations in primary sediment sources, standard deviation (SD), result of the Kruskal–Wallis test and Discriminant Function Analysis (DFA), used to identify the ideal set of the discrimination of sources that supply sediment to the main reservoir (MR).

| Fingerprinting Property | Sediment Sources | | | | | | Kruskal–Wallis Test | | DFA | | |
|---|---|---|---|---|---|---|---|---|---|---|---|
| | Bare Soil (n = 23) | | Riverbank (n = 20) | | Bed (n = 8) | | *H*-Value | *p*-Value | Wilks's Lambda | Partial Lambda | *p*-Value |
| | Mean | SD | Mean | SD | Mean | SD | | | | | |
| Pb (mg kg$^{-1}$) | 22.59 | 8.13 | 22.10 | 11.53 | 18.90 | 12.15 | 1.71 | 0.42 * | - | - | - |
| Cr (mg kg$^{-1}$) | 26.71 | 21.45 | 5.64 | 7.60 | 18.02 | 17.24 | 19.50 | 0.00 | 0.40 | 0.80 | 0.01 |
| Cu (mg kg$^{-1}$) | 78.39 | 38.97 | 74.70 | 54.90 | 83.76 | 72.60 | 1.03 | 0.60 * | - | - | - |
| Mn (g kg$^{-1}$) | 0.26 | 0.16 | 0.10 | 0.05 | 0.24 | 0.20 | 16.74 | 0.00 | 0.39 | 0.81 | 0.01 |
| Ni (mg kg$^{-1}$) | 23.6 | 13.27 | 14.38 | 12.73 | 15.14 | 16.27 | 7.56 | 0.02 | 0.33 | 0.95 | 0.34 * |
| Mg (g kg$^{-1}$) | 0.46 | 1.31 | 0.32 | 0.18 | 5.55 | 14.08 | 12.16 | 0.02 | 0.33 | 0.95 | 0.35 * |
| Zn (mg kg$^{-1}$) | 37.85 | 17.78 | 73.43 | 73.92 | 68.01 | 69.50 | 1.65 | 0.44 * | - | - | - |
| Fe (g kg$^{-1}$) | 62.12 | 32.28 | 32.01 | 24.34 | 47.04 | 42.17 | 9.32 | 0.01 | 0.33 | 0.95 | 0.33 * |
| Al (g kg$^{-1}$) | 24.96 | 13.55 | 18.92 | 12.63 | 5.80 | 5.86 | 14.78 | 0.00 | 0.45 | 0.71 | 0.00 |
| Cd (mg kg$^{-1}$) | 0.94 | 0.64 | 0.56 | 0.42 | 0.76 | 0.68 | 4.98 | 0.08 * | - | - | - |

* Not significant at $p < 0.05$ level.

The second statistical test (DFA), based on the minimization of the lambda, was applied to the metals that passed the first test. The DFA indicated which of the metals selected in the first test maximized source identification. In this test, only three of the six initially selected metals were considered as the ideal set to maximize source discrimination (Table 1), as they presented a *p*-value less than or equal to the significance of 5%. As an ideal set, the metals Cr, Al, and Mn were selected, with the ability to correctly identify the origin of the sediment by the set of metals with 86% accuracy.

To assess the characteristics of the sediments deposited in the reservoir, we carried out sediment monitoring with suspended sediment collections of the Cabaça (C) and Bandeira (B) streams, before discharge into the lake. Suspended sediment collections occurred after rain events with the capacity to mobilize material. We carried out five collection campaigns in each tributary stream, totaling 10 samples. The collections from the sampler installed in the Bandeira (B) stream were named from B1 to B5, and the collections from the sampler installed in the Cabaça (C) stream were named from C1 to C5 (Table 2).

**Table 2.** Collection of sediment samples in integrated time suspension: B samples from the sampler installed in the Bandeira (B) stream and C samples from the sampler installed in the Cabaça (C) stream.

| Sample | Date | Sample Period | Weight (g) | Accumulated Rainfall (mm) |
|--------|------|---------------|------------|---------------------------|
| B1 | 27 September 2019 | 46 days | 9.22 | 17.80 |
| C1 | | | 81.93 | |
| B2 | 31 October 2019 | 34 days | 577.99 | 30.80 |
| C2 | | | 762.90 | |
| B3 | 23 December 2019 | 10 days | 405.69 | 71.20 |
| C3 | | | 178.29 | |
| B4 | 28 January 2020 | 36 days | 2030.16 | 334.00 |
| C4 | | | 965.79 | |
| B5 | 6 March 2020 | 38 days | 1573.24 | 253.80 |
| C5 | | | 899.13 | |

The sediment samples showed variable characteristics and quantities of material throughout the samplings. The sediment transported by the Bandeira (B) stream mainly consists of medium-grained sand. Meanwhile, the material carried by the Cabaça (C) stream is formed by medium and fine sand and boulders. However, the amount of rain interfered with the granulometry of the material. The B1 and C1 collections took place in a period of little rain in the basin, with an accumulated rain of 17.80 mm, resulting in finer sediments in a smaller quantity. Collections B4, C4, B5 and C5 occurred in periods of intense rain, with an accumulated rainfall of 344.00 mm and 253.80 mm, respectively, and carrying material in larger quantities and with larger diameters (Figure 3).

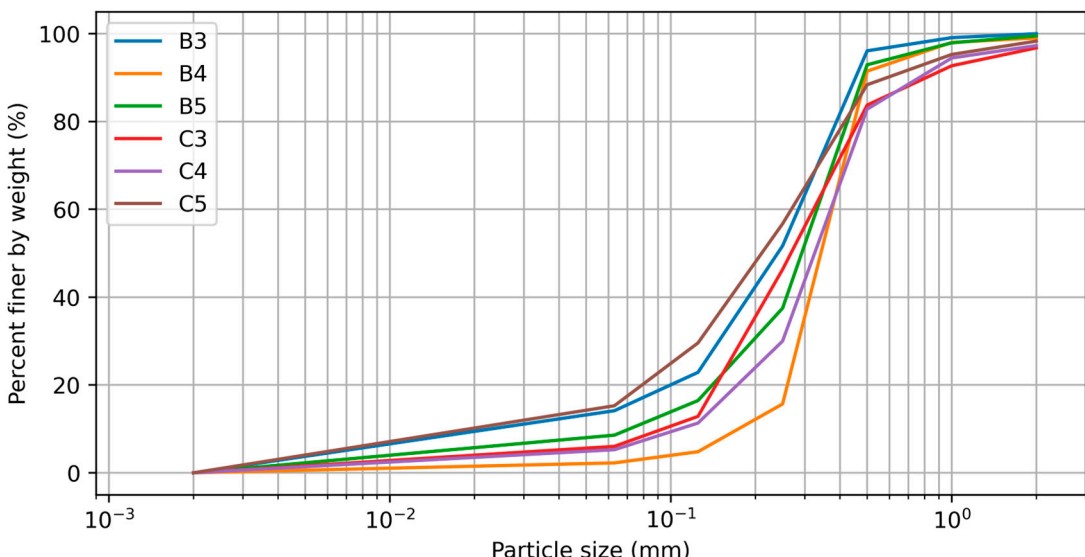

**Figure 3.** Particle size composition of suspended sediment samples from Bandeira (B) and Cabaça (C) streams.

We verified, by chemical analysis, the presence of all metals in the suspended sediment samples (Table 3). The metal concentrations of the Bandeira (B) stream and the Cabaça (C) stream did not present significant differences when compared using the Kruskal–Wallis test. There were also no significant differences in metal concentrations between one sample and another, by the same test.

**Table 3.** Metal concentration in suspended sediment samples collected in the Bandeira (B) stream and Cabaça (C) stream, which contribute to the main reservoir.

| Fingerprinting Property | Bandeira (B) (*n* = 5) | | Cabaça (C) (*n* = 5) | |
|---|---|---|---|---|
| | Mean | SD | Mean | SD |
| Pb (mg kg$^{-1}$) | 32.92 | 8.54 | 35.63 | 5.60 |
| Cr (mg kg$^{-1}$) | 21.05 | 6.55 | 17.89 | 7.00 |
| Cu (mg kg$^{-1}$) | 127.09 | 44.17 | 113.67 | 36.15 |
| Mn (mg kg$^{-1}$) | 330.77 | 170.92 | 185.29 | 110.73 |
| Ni (mg kg$^{-1}$) | 29.56 | 12.89 | 23.66 | 9.38 |
| Mg (g kg$^{-1}$) | 1.40 | 0.41 | 1.41 | 0.27 |
| Zn (mg kg$^{-1}$) | 240.77 | 81.99 | 303.24 | 220.91 |
| Fe (g kg$^{-1}$) | 59.26 | 19.14 | 63.55 | 13.00 |
| Al (g kg$^{-1}$) | 37.95 | 11.51 | 28.33 | 5.59 |
| Cd (mg kg$^{-1}$) | 0.22 | 0.49 | 0.27 | 0.60 |

The origin of the sediment samples from the two catchments was identified by the combination of Cr, Al and Mn metals applied in the multivariate model, based on the metal concentrations of the three primary sources (Table 1), and suspended sediment samples (Table 3). From the 10 suspended sediment samples, three had an MAE above 20%, which is considered the method limit (Table 4). These samples were disregarded, as the model was not able to correctly discriminate the origin of the sediment.

**Table 4.** Relative contribution of sediment-producing sources to the main reservoir.

| Sample | Bare Soil (%) | Bed (%) | Riverbank (%) | Error (%) |
|---|---|---|---|---|
| B1 | 10 | 10 | 97 | 40 [a] |
| B2 | 31 | 43 | 26 | 0.00 |
| B3 | 72 | 19 | 9 | 0.00 |
| B4 | 80 | 14 | 6 | 0.00 |
| B5 | 90 | 10 | 0 | 22 [a] |
| C1 | 63 | 17 | 20 | 0.00 |
| C2 | 22 | 8 | 70 | 0.00 |
| C3 | 30 | 9 | 61 | 0.00 |
| C4 | 25 | 9 | 66 | 0.00 |
| C5 | 96 | 5 | 0 | 25 [a] |

[a] Not correctly classified due to Mean Absolute Error above 20%.

Only in samples B2, B3, and B4 of the five sediment samples from the Bandeira (B) stream could the origin of the sediments be identified. In B2, the main sediment contribution was from the bed, and the smallest was from the banks. Meanwhile, in B3 and B4 there was an increase in the contribution of bare soil, which became the main source. Bank material continued contribute the least. From the sediment samples of the Cabaça (C) stream, in C1, C2, C3, and C4, the sources were discriminated. In C1, the bare soil source was the main contributor, and the bed sediment was the one with the smallest proportion. In C2, C3, and C4, there was a change in the origin of the sources, and the bank became the main producer of the sediments carried by the stream. In general, the sediment transported by the Bandeira (B) stream comes from bare soils, and the Cabaça (C) stream comes from the bank (Figure 4).

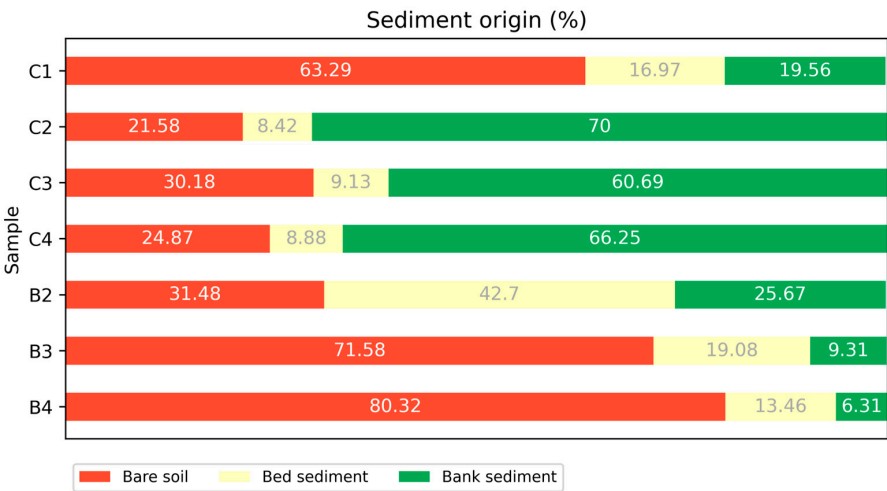

**Figure 4.** Relative contribution of the Bare soil, Bed and Bank sediment-producing sources to the main reservoir tributaries, Bandeira (B) stream and Cabaça (C) stream.

The classification of land use and land cover (Figure 5) indicated that, in the drainage basin, delimited as an area of contribution to the lake, the urbanized area increased from 49.6% to 52.1%. The area of undergrowth ranged from 18.6% to 21.3%, and the area of arboreal vegetation ranged from 12.5% to 13.6% between 2011 and 2019. The bare soil class, in the same period, showed a decline, reducing from 19.2% to 12.8%. The classes with the greatest variations during the period were undergrowth and bare soil.

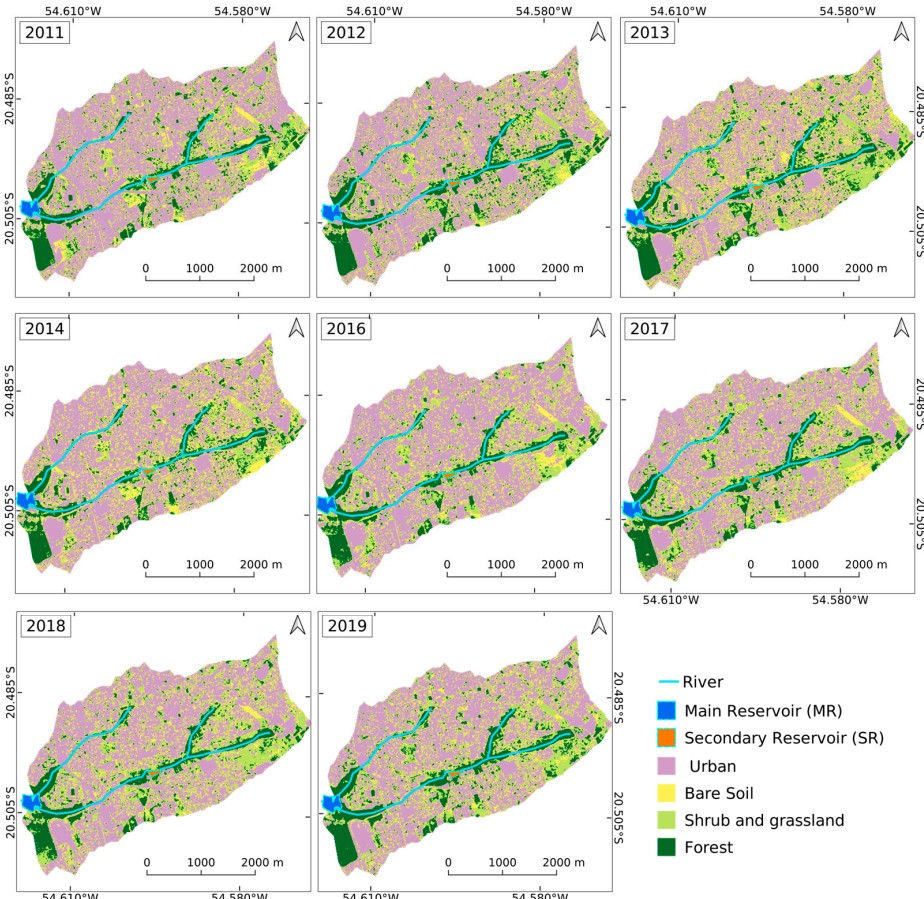

**Figure 5.** Classification of land use and land cover in the analyzed period (2011–2019), considering the use classes: urban, forest, shrub and grassland, and bare soil.

At the beginning of the suspended sediment sampling (2019), the drainage basin, according to the classification of use and occupation, comprised 52% impermeable area, 13% bare soil, 21% undergrowth, and 14% arboreal vegetation. The origins of sediments and the contribution of each source are due to this occupation characteristic.

We verified the correlation between land use and land cover variables for the evaluated years (Figure 6). The analysis indicated a significant negative correlation between bare soil and undergrowth (r = −0.919 *p* = 0.001). The other variables showed no correlation.

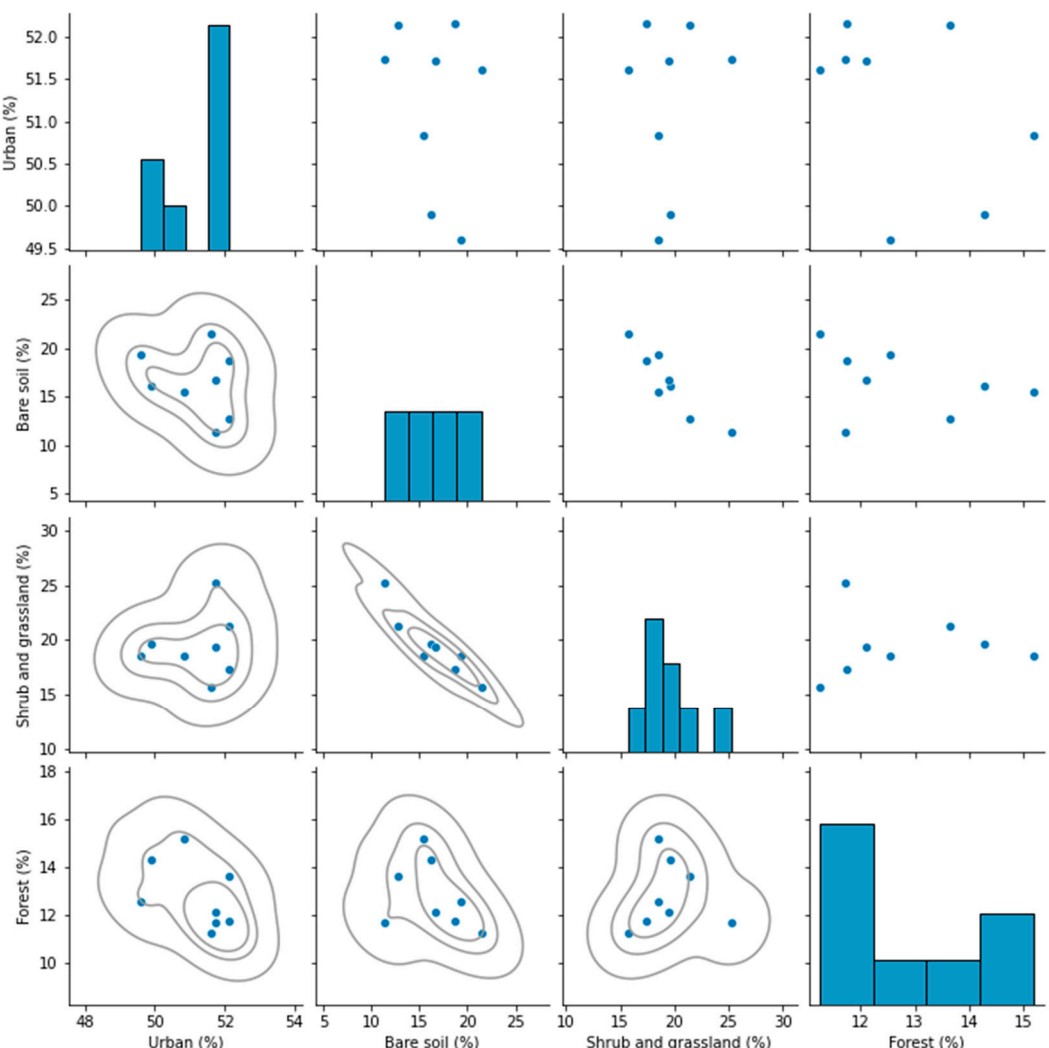

**Figure 6.** Scatter plots for land use and cover distribution over 8 years (2011–2019).

## 4. Discussion

We found an intensification in the siltation of the main reservoir over the years. The periods that showed the greatest simultaneous reductions in area and volume, respectively, were 2016–2017 (6.35% and 8.80%) and 2017–2018 (17.38% and 7.63%). Data for 2008 were considered as the base year, due to the difference in methodology adopted in bathymetric surveys.

In tracing the sources, samples B1, B5, and C5 were discarded as they presented a MAE greater than 20%, indicating that the model was not able to discriminate the origin of the sediments in these samples. The B1 collection took place after a long period of drought, a characteristic of the region, and the first rains mobilized the sediments deposited in the bed and in the basin during the drought. The last sample collection (B5 and C5) occurred after the most intense rain period in the basin. The sediments collected in samples B1, B5, and C5 had the highest concentrations of metals Pb, Cu, Zn and Cd, which have a

strong relationship with urban pollution [27,37–39]. Despite the metals being found in the basin's soil samples, the high concentrations found in the sediment are evidence that diffuse pollution possibly contribute to it.

The error in discriminating the sources in samples B1, B5, and C5 indicates that the SSF method is not able to determine the origin of sediments collected after long periods of drought and after periods of intense rain. In long periods of drought, the first rains carry the pollution deposited in the soil to water courses, a phenomenon known as the first flush; this interferes with the concentration of metals due to the carrying of anthropogenic metals from the basin soil. In periods of intense rain, the increase in the speed and intensity of runoff in urban catchments causes the resuspension of previously deposited sediments to occur, interfering with sampling.

This can be explained as the urban basin is the main producer of anthropogenic contaminants [40], and in urban rivers the increase in metals from sewage and other pollution makes it difficult to identify sources. The transport of mixtures of new and old sediments in rivers with unnatural beds, dams, and other hydraulic interferences [41] can also interfere with determining the origin of the sediment. Peter et al. [42] identified that, in addition to carrying contaminants related to the first flush in urban basins, there is an increase in pollutants with the increase in runoff, as more distant sources are mobilized. Therefore, in studies of this kind, it is essential to observe the basin's rainfall history to carry out sediment sampling.

For the collections designated as B2, B3, B4 and C1, C2, C3, and C4, the origin of the sediments by the model could be determined. During the sampling period, the pollutants had already been washed away by the first rains. Thus, there was no interference in metal concentrations, and precipitation events occurred within the average.

Despite the increase in the loss of volume and area of the reservoir in recent years (2016–2018), the changes in land use and occupation were not accentuated in the basin. In the period of the greatest loss of area and volume (2016 to 2018), the urbanized area varied 0.13% between 2016 and 2017, and did not vary from 2017 to 2018. The area of arboreal vegetation increased by 0.45% in the period of 2016 to 2017, and maintained the increase in the area at 0.41% between 2017 and 2018. The basin area classified as undergrowth changed with an increase in area of 9.52% between 2016 and 2017 and a decrease of 5.81% in the period from 2017 to 2018. Areas of bare soil followed an opposite trend of undergrowth areas. We found that between 2016 and 2017, these areas showed a decline of 10.10%, and between 2017 and 2018, an increase of 5.42%.

Variations in bare soil and undergrowth are inversely proportional in the temporal analysis of use and occupation of the basin, possibly due to the effect of the amount of rainfall. If the dry period is characterized as severe, grasses do not develop, leaving the soil without plant cover. If there is enough rain, the plants will grow by covering the ground. We also noted that variations occurred in the arboreal class. We found an increase in the arboreal area between 2011 and 2013, a decrease between 2013 and 2016 and, in the last years analyzed (2017–2019), we verified an increase again. The classification of the images indicated that the areas on the banks of the streams, which have arboreal vegetation, are still preserved without suffering vegetative loss, and that the sources of the Bandeira (B) stream, which are protected with by a permanent preservation area (PPA), under recovery and fenced, contributed to the increase in the arboreal vegetation in the 2017 and 2019 periods.

The sediment monitoring that took place in 2019 and the tracing of sources indicated that the Bandeira (B) stream transports greater amounts of sediment compared to the Cabaça (C) stream. The origins of the sediments in the two catchments differ, despite having the same type of soil. Studying urban lakes, Franz et al. (2014) identified that the main contribution of urban areas to silting was the bare soils found in expanding areas. The catchment of the Bandeira (B) stream is undergoing a process of urban densification, where there are empty spaces, paved and unpaved streets. In this catchment, the main

sources of sediments are bare soils, which contribute, on average, to 61% of the sediment transported to the lake.

The banks have the smallest contribution in the Bandeira (B) basin, with an average of 14%. The low contribution of the bank source to the sediments is due to the vegetative protection of the margins [43,44], as there is a PPA recovery project in its springs, and most of the margins have developed trees. Another justification would be the sealing of the channel upstream of the collection point, as was also reported by Vercruysse and Grabowski [45], which reduced the sediment production from these sources.

The main origin of the sediments carried by the Cabaça (C) stream is the riverbank, contributing 54% on average. The Cabaça (C) stream basin is densely occupied, with few empty spaces and completely paved streets. In urban areas with dense occupation and paving, riverbanks are the main sources of sediments. Carter et al. [18] found riverbank sediment values ranging from 43 to 84%. For [28], the average contribution of the riverbanks was 58% and for Cashman et al. [46], 87% of the average proportion of suspended sediment was composed of the riverbank material. The contribution of the bare soil class represents 35% and is the second source that contributes the most to the sediments of the Cabaça (C) stream.

The bed material contributes more to the sediments carried by the Bandeira (B) stream, representing 25% of the total, than to the sediment of the Cabaça (C) stream, where it contributes with 11%. This is possibly a result of the characteristics of the beds, which, in Bandeira (B), are sandy with the formation of sand banks; these are unstable and are easily carried by the flow. The bed of the Cabaça (C) is clay and eroded due to the intense flow. Thus, it presents stability and resistance to erosive flow, as it is made of material rich in clay and iron oxides [44], and hinders the accumulation of material. Bank and bed erosion is a significant source of sediment contribution in urban areas [47]. The remobilization of sediment deposited in the channel bed plays a fundamental role in urbanized basins, modifying the sediment rates downstream [48].

Changes in land use and land cover occurred mainly in the Bandeira (B) stream basin, since in the occupation process, the streets are being paved and the land is becoming impermeable. On the other hand, the drainage area of the Cabaça (C) stream is urbanized, with paved roads and impermeable soil, justifying the few changes in land cover. In the basin undergoing occupation, the main sources of sediment come from bare soils [19]. The basin is fully occupied, and the main sediment contributions come from the erosion of the banks and the bottom of the channels [49]; this was also identified in this work.

The classification of land use and occupation indicated few changes in the basin during the period analyzed (2011–2019), and did not justify the intensification of silting in the reservoir. With minimal variations in the use and occupation of the basin over the years, we assume that the percentage of contribution from sources has changed little. This finding led us to analyze other factors that may have contributed to silting and that were not identified in the temporal analysis of the basin's occupation.

We verified that in 2012, there were interventions in the bed of the Bandeira (B) stream to correct an accelerated erosion process in the surroundings of the secondary reservoir. During this period, the Bandeira (B) stream was channeled to prevent further erosion. In 2014, the secondary reservoir was completely silted, which caused the sediment that was previously deposited there to now to be deposited downstream, reaching the main reservoir. In 2017, there was an intervention in the bed of the Cabaça (C) stream to correct the erosion of the banks after a retaining wall was installed.

In addition to human interventions in the beds, another factor is the interconnection of the drainage network with water courses. In the analyzed basin, streams and lakes are part of the drainage network of the urbanized area of the municipality, and the main reservoir works as a containment basin for outflows. The region has experienced flooding problems, and the lake naturally retains much of the sediment carried by the streams. The high connectivity of the drainage network with watercourses in urban basins allows sediment particles to be more easily transported to watercourses [50]. With the advance of occupation

in the Bandeira (B) hydrographic basin, the roads are being paved and receiving a drainage system. Thus, the interconnection of urbanized areas with the drainage network is essential to indicate the urban impact on the production of sediments.

The siltation of urban reservoirs is a result of land use and land cover. The tendency is that, over time, with the advance of occupation and waterproofing, the sediment rates will decrease until they stabilize. Although urban waterproofing blocks some sources, other sources are inserted, as reported by Russell et al. [51]. Gellis et al. [52] found the highest rates of sediment production in an urbanized basin in the 1980s, indicating that, even after the occupation process was completed, the basin continued to produce sediment, due to the erosion of the banks and channels. Russell et al. [53], in turn, questioned the concept that the established urban coverage reduces the production of sediments, as in their study, they found that a significant proportion of sediment had originated from an urbanized area for over 40 years.

Despite the use and occupation of the basin at different times and the different hydraulic characteristics, the two sub-basins contribute to the siltation of the main reservoir; this is made clear by assessing the formation of a sand bank in the flow of streams into the reservoir. It is possible that interventions into the bed and the silting of the secondary reservoir contributed more intensively to the mobilization and remobilization of sediments. These actions increased the deposition of sediments, with a consequent siltation process in the reservoir. The sediments that are carried to the main reservoir, in total, come from the producing sources in the following proportions: Bare soil, 46.9%, Bank, 37.10%, and Bed, 17.2%.

Our results show that the silting process of urban reservoirs is not only associated with changes in land use and occupation, but with direct interferences in water courses that possess potential for sediment mobilization; however, they are not considered in the assessments of the origin of sediments as they are transitory contributions. Work on channels and banks need proper management to avoid sediment mobilization.

In addition, there is a temporal lack between all analyzed datasets: bathymetric data (collected in 2008–2018), sedimentological data (collected in 2019–2020), and LULC (analyses for 2011–2019). Thus, the reservoirs siltation might have accumulated sediments from past events since they were built in the 1970s. The period between 1970 and 2008 (almost 40 years) comprises many LULC changes in the catchment area with the progress of urbanization; however, the lack of information about the reservoir's initial conditions (capacity) does not allow us to draw any conclusion about this period. However, as the studied catchment started its urbanization (LULC changes) at the 1970s [25], the period of our study (recent years) is a sample of what could have happened over the past years (1970s–2008), concerning the sediment dynamics.

## 5. Conclusions

The silting process is intensified in hydrographic basins, even with minimal variations in the use and occupation of the basin's soil. There is a proportional relationship between the reduction in the reservoir volume and the increase in impermeable areas in the basin. Rainfall directly interferes in the production of sediments; if there is regular rain in the basin, the bare soils are covered by grasses and the contribution of this source is reduced in the silting process.

The use of bathymetric surveys and data on land use and occupation, associated with the source tracing technique, are alternatives to identifying sediment mobility in urban basins, especially in those where the drainage network is connected to water courses. The temporal evaluation of land use and land cover alone is not sufficient to indicate a variation in the sediment-producing areas. The same happens if we evaluate only the bathymetric surveys. By associating the bathymetric monitoring of the reservoir with the temporal evaluation of land use, land cover, and the tracing of sources, we can affirm that the production of sediment is linked to other factors, as well as to changes in land cover.

In urban basins, the factors that most contribute to the silting up of reservoirs are the erosion of banks and beds, sediment remobilization and the connectivity of the drainage network with water courses. The interconnection of drainage to streams affects sediment transport and alters the hydro-sedimentological characteristics of the channels.

Continuous hydro-sedimentological monitoring allows for more consistent conclusions regarding the impact of the use and occupation of the basin on the production of sediments. The urbanization process is the variable that most impacts the production of sediments. Thus, when evaluating the production of sediments in urbanized basins, we must consider the impermeability of the soil, the increase in surface runoff, the increasing density of drainage networks, and modifications and works in the canal bed.

**Author Contributions:** M.E.A.F., P.T.S.O., T.A.S. and P.T.S.O. conceived the ideas and designed the methodology for the study. M.E.A.F. and G.A.C. performed field work and collected data. M.E.A.F., G.A.C., J.A.A.A. and P.T.S.O. processed, and analyzed the data. M.E.A.F., D.A.Z., J.A.A.A. and T.A.S. led the writing of the manuscript. All authors contributed to the final manuscript revision for peer-review submission. All authors have read and agreed to the published version of the manuscript.

**Funding:** This research was funded by Ministry of Science, Technology, Innovation and Communication—MCTIC and the National Council for Scientific and Technological Development—CNPq, grant numbers 303128/2018-6, 422947/2018-0, 309752/2020-5. The graduate students were supported by the Coordination of Improvement of Higher Education Personnel—CAPES (finance code 001).

**Institutional Review Board Statement:** Not applicable.

**Informed Consent Statement:** Not applicable.

**Data Availability Statement:** Not applicable.

**Acknowledgments:** The authors acknowledge the Graduate Program in Environmental Technologies—PPGTA (UFMS-FAENG) for the scientific support. The authors would like also to thank the editor and the anonymous referees for their useful comments, which substantially improved the manuscript.

**Conflicts of Interest:** The authors declare no conflict of interest.

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
