# Peer review of "Fingerprinting Sediment Origin of the Silting Process of Urban Reservoirs"

_sustainability, doi:10.3390/su15031745_

Round 1

Reviewer 1 Report

This research article was well edited based on authors's original research results. Since there were some problems, this research article need to revise before the publication as one of research articles in international journal.

Please refer followings.

[Major reversion]

1. Data: Particle size distribution

If possible, please use the data of particle size distribution at each sediment sample in this research article.

[Minor reversion]

2. Keywords

If possible, please use at least 5 important terms for keywords.

3. Effective numbers

Please use same effective numbers in a same table or research article.

Please use at least 3 or more effective numbers(EFNs) in Table 1, Table 2 and Table 3.

Ex.) EFNs 3: 22.6, 8.13, 22.1, 1.71, 0.400

      EFNs 4: 22.59, 8.130, 22.10, 1.710, 0.4000

Author Response

Reviewer # 1

This research article was well edited based on authors's original research results. Since there were some problems, this research article need to revise before the publication as one of research articles in international journal.

Please refer followings.

Dear Colleague,

Thank you for all your valuable suggestions. They helped us to markedly improve the presentation of the study.

[Major reversion]

  1. Data: Particle size distribution

If possible, please use the data of particle size distribution at each sediment sample in this research article.

Response: We show the particle size distribution for each sediment sample on Figure 3 (lines 298-300).

  1. Keywords

If possible, please use at least 5 important terms for keywords.

Response: Thank you for this remark, we included a 5th keyword to complete the requested 5 important terms (lines 30-31).

  1. Effective numbers

Please use same effective numbers in a same table or research article.

Please use at least 3 or more effective numbers (EFNs) in Table 1, Table 2 and Table 3.

Ex.) EFNs 3: 22.6, 8.13, 22.1, 1.71, 0.400

      EFNs 4: 22.59, 8.130, 22.10, 1.710, 0.4000

Response: Thank you for your suggestion. We carefully checked our data sources and we do not have enough precision from our analytical methods to use more than 2 EFNs. Thus, we carefully checked our tables and standardized the EFNs on each table.

Reviewer 2 Report

Dear Authors,

the paper presents the research on the functioning of dammed Campo Grande catchment. The topic can be considered interesting and original, but the manuscript needs a minor remark before the next step of submission. The results and discussion are presented well, but the methodology part needs to provide more detailed information – see listed below. Also, please consider providing (i) hydrological data (if exist), and (ii) a discussion about the background that lacks your research in terms of hydrological analyses, lack of sedimentological data from the 60s to 2008, and lifetime formulas – all see listed below.

General remarks:

Please add information about the elevation of the study area catchment; detailed information about reservoirs (MR and SR), such as initial volume, capacity, max depth, etc. Also, please add general information about climate conditions or the catchment (precipitation, etc.) it’s important information about sediment dynamics analyses.

Please provide detailed information about the methodology of “bathymetric gaugings”, you use boat? Also, please provide detailed information about “acoustic profiler”, what kind of equipment you use? What kind of software you use to pre-processing of bathymetric data? This information is missing.

Also, inline 187 you wrote that “maximum depth found was 4.89 m” – please provide information about accuracy of bathymetric measurements, if you show “2 decimal pleases” it means that you use geodetic method to have “cm +/- scale”, please provide more information to Methodology part.

What kind of software you use to perform the DEM of the bathymetry? Please provide detailed information to the methodology. Next, why you use “Nearest Neighbor geostatistical method” – please explain? Also, you wrote “obtained the volume and area of the reservoir” please provide information about the reference water level? Is there any information about water level operational?

Please provide information, if that was any flushing of sediments from the reservoir in the past? Also, that you have information about cleaning it’s?

Please provide information about the source of RapidEye and Sentinel-2 images – this information is missing; and some characteristics of spatial data that you use. Also, provide information about pre- and post-processing of spatial data (software, etc.), what kind of scale of px that you have, enough? Please provide information; next what kind of formulas and software that you use to determine LULC?

I have three general remarks, that need to be explained in a manuscript or pointed out that your research has these lacks:

(i)              Your research mainly presented quality and quantity analyses of sediment dynamics in the years 2008-2018, based on bathymetric data (collected in 2008-2018), sedimentological data (collected in 2019-2020), LULC (analyses for 2011-2019) – so you have temporal-lack; your fingerprint results (especially based on sediment samples) showed also accumulation/erosion conditions from an earlier period (start from 70s) when reservoirs were created; please provide information how the influence of sedimentological conditions from 70’s to 2008 affect on your results?

(ii)            Sediment transport is strongly related to the hydrological conditions of the catchment, but in your manuscript hydrological data/analyses/modeling is missing! It’s a strong lack of research – please provide information about hydrological conditions or give information about it.

(iii)           Many formulas described the siltation of reservoirs, e.g. Gill method, Brune method, Goncarov’s method, Stonawski formula, Hartung method, etc. You’re not described and discussed your results with mentioned methods – please consider, this because in the analysis period (2008-2018) you have a very intensive capacity reduction (45%) and we don’t know what was initial capacity? (at the 60s)? We don’t know what was the reduction rate from the beginning? And most important how do your results compare with the lifetime useful of reservoir criteria eg. described by Hartung?

I add some comments on different lines of the document.

L14: “tributary bodies” – I suggest deleting “bodies”; tributary as creak/stream / river is enough to describe

L70 “of channel” – I suggest deleting

L130: “particle size distribution” please provide reference to a method of PSD determination, USGS, or some different?

L181: “volume” is not correct -> capacity

L181: “water mirror” is not correct – water or area

Best regards

Author Response

Reviewer #2

Dear Authors,

the paper presents the research on the functioning of dammed Campo Grande catchment. The topic can be considered interesting and original, but the manuscript needs a minor remark before the next step of submission. The results and discussion are presented well, but the methodology part needs to provide more detailed information – see listed below. Also, please consider providing (i) hydrological data (if exist), and (ii) a discussion about the background that lacks your research in terms of hydrological analyses, lack of sedimentological data from the 60s to 2008, and lifetime formulas – all see listed below.

Dear Colleague,

Thank you for your appointments. Your comments were indeed very helpful in improving our manuscript. We have performed all the required changes and answered all questions. We appreciate all the given suggestions and hope that our revision will now meet your suggestions.

General remarks:

Please add information about the elevation of the study area catchment; detailed information about reservoirs (MR and SR), such as initial volume, capacity, max depth, etc. Also, please add general information about climate conditions or the catchment (precipitation, etc.) it’s important information about sediment dynamics analyses.

Response: Thank you for this important remark. We added all available information about the study area (lines 80-104). Initial volume, capacity, depth, etc. from the reservoirs are not available to be added in this study.

Please provide detailed information about the methodology of “bathymetric gaugings”, you use boat? Also, please provide detailed information about “acoustic profiler”, what kind of equipment you use? What kind of software you use to pre-processing of bathymetric data? This information is missing.

Response: Thank you to remember us about missing information (line 118). We added pieced of text including all requested information.

Also, inline 187 you wrote that “maximum depth found was 4.89 m” – please provide information about accuracy of bathymetric measurements, if you show “2 decimal pleases” it means that you use geodetic method to have “cm +/- scale”, please provide more information to Methodology part.

Response: Depth measurements have 1% of accuracy and this information is available on section 2.2 (line 121).

What kind of software you use to perform the DEM of the bathymetry? Please provide detailed information to the methodology. Next, why you use “Nearest Neighbor geostatistical method” – please explain? Also, you wrote “obtained the volume and area of the reservoir” please provide information about the reference water level? Is there any information about water level operational?

Response: Thank you for asking about volume and area calculation details. We added useful information on section 2.2. to address this comment (lines 124-133).

Please provide information, if that was any flushing of sediments from the reservoir in the past? Also, that you have information about cleaning it’s?

Response: The reservoirs do not have a hydraulic structure in the dam that would allow a sediment flushing. We added in the study site description (section 2.1) that there was no sediment flushing (lines 97-104).

Please provide information about the source of RapidEye and Sentinel-2 images – this information is missing; and some characteristics of spatial data that you use. Also, provide information about pre- and post-processing of spatial data (software, etc.), what kind of scale of px that you have, enough? Please provide information; next what kind of formulas and software that you use to determine LULC?

Response: We included all requested information on section 2.4 (lines 206-217). Thank you for suggesting this improvement. Important remark: Sentinel-2 imagery are currently not available on USGS repository, but at the time of the development of this manuscript, the data could be downloaded there.

I have three general remarks, that need to be explained in a manuscript or pointed out that your research has these lacks:

(i)    Your research mainly presented quality and quantity analyses of sediment dynamics in the years 2008-2018, based on bathymetric data (collected in 2008-2018), sedimentological data (collected in 2019-2020), LULC (analyses for 2011-2019) – so you have temporal-lack; your fingerprint results (especially based on sediment samples) showed also accumulation/erosion conditions from an earlier period (start from 70s) when reservoirs were created; please provide information how the influence of sedimentological conditions from 70’s to 2008 affect on your results?

Response: Thank you for this important remark. As we do not have any data between 1970 and 2008, we cannot draw any conclusion. Thus, we added a piece of discussion addressing this limitation of our study (lines 497-506).

(ii)            Sediment transport is strongly related to the hydrological conditions of the catchment, but in your manuscript hydrological data/analyses/modeling is missing! It’s a strong lack of research – please provide information about hydrological conditions or give information about it.

Response: We added information about hydrometeorological conditions on section 2.1. to address the study site context (lines 80-93), however, it is an ungauged basin and we do not have on-site hydrological data (rainfall and flow rate) for the study period.

(iii)           Many formulas described the siltation of reservoirs, e.g. Gill method, Brune method, Goncarov’s method, Stonawski formula, Hartung method, etc. You’re not described and discussed your results with mentioned methods – please consider, this because in the analysis period (2008-2018) you have a very intensive capacity reduction (45%) and we don’t know what was initial capacity? (at the 60s)? We don’t know what was the reduction rate from the beginning? And most important how do your results compare with the lifetime useful of reservoir criteria eg. described by Hartung?

Response: Thank you for your suggestion. The main idea of this manuscript was to identify potential sources of sediments that contributed to the siltation of a reservoir located in the outlet of an urban catchment. In addition, we do not have information about the reservoir initial capacity to completely address your suggestion. However, your comment is a very nice idea for a future study in the area, and helped us to address an adjustment in the purpose of our study and make it more suitable to the methods and presented results, discussion and conclusions. The adjusted purpose of our study is: “Thus, we characterize the potential sources of sediment responsible for the siltation process in urban reservoirs, considering the changes in land use and occupation to identify possible drivers that might contribute to silting”. Thus, the reservoir siltation is not the focus of the manuscript anymore. The bathymetry timeseries and observed changes (siltation) along the monitoring period (2008-2018) is evidence of the impact of land use changes observed for the same period, with an increase of urban areas along the years, connecting the streams to bare soil areas, which were identified by SSF as major sources of sediment found in the samplers installed before the main reservoir.

I add some comments on different lines of the document.

L14: “tributary bodies” – I suggest deleting “bodies”; tributary as creak/stream / river is enough to describe

Response: We deleted the term. Thank you for your suggestion.

L70 “of channel” – I suggest deleting

Response: We deleted the term. Thank you for your suggestion.

L130: “particle size distribution” please provide reference to a method of PSD determination, USGS, or some different?

Response: Thank you for remarking this issue. We added the PSD determination method using standard mesh sizes (line 172).

L181: “volume” is not correct -> capacity

Response: Thank you for correcting this term. We changed it in the text.

L181: “water mirror” is not correct – water or area

Response: Thank you for correcting this term. We changed it in the text.

 Best regards

Reviewer 3 Report

My comments about the paper are as follow:

1.      Obtained results should be clearly stated in the abstract.

2.      The introduction is poorly written and in some cases unclear. It needs to be rewritten in a clear manner, and the novelty and the overall contributions should be clearly stated. What are the motivations of the study? What is new in comparison to what is already done?

3.      More details about the section 2.1.

4.      Section 2.2. This section is informative however, it needs to be improved: 08 bathymetric gauging’s stations are selected, therefore, the coordinates of each one should be provided and also the total number of point. It is more suitable if the authors provide a location carte for the bathymetric gauging’s stations.

5.      Section 2.4 is unclear ???

6.      Figure 2 is for bad quality.

7.      More details about the information’s reported in Table 1 is necessary for improving our understanding of the paper.  

8.      I wonder if the authors can improve Figure 5?

9.      All the remaining parts of the paper are good, I am sure that this is a good paper.

Author Response

Reviewer #3

My comments about the paper are as follow:

Dear Colleague,

Thank you for all your valuable suggestions. They helped us to markedly improve the presentation of the study.

  1. Obtained results should be clearly stated in the abstract.

Response: We added more information about the results in the abstract. Thank you for your suggestion (lines 10-29).

  1. The introduction is poorly written and in some cases unclear. It needs to be rewritten in a clear manner, and the novelty and the overall contributions should be clearly stated. What are the motivations of the study? What is new in comparison to what is already done?

Response: We added extra pieces of information in the Introduction to address this comment (lines 34-77). We also changed some parts of the introduction to make it clearer.

  1. More details about the section 2.1.

Response: We added more information about the study site context in order to address this comment (lines 80-104). Thank you for this suggestion.

  1. Section 2.2. This section is informative however, it needs to be improved: 08 bathymetric gauging’s stations are selected, therefore, the coordinates of each one should be provided and also the total number of point. It is more suitable if the authors provide a location carte for the bathymetric gauging’s stations.

Response: Thank you for asking about this issue. We changed the term “gaugings” to “surveys”, as it was a procedure done using a moving boat to assess the bathymetry of the reservoir once a year along the monitoring period. The location of the lake (MR) is given by Figure 1. Additionally, the number of points depends of the area of the lake, which varied along the series. Thus, we opted to state the density of points that was set to the acoustic surveyor (0.05 points/m²). Extra information were added to this section to address this comment (lines 113-133).

  1. Section 2.4 is unclear ???

Response: We added more information about imagery classification and data characteristics (lines 206-223).

  1. Figure 2 is for bad quality.

Response: We improved figure 2 (lines 243-245): we changed its layout, changed the font size and also standardized the style in accordance with the other figures.

  1. More details about the information’s reported in Table 1 is necessary for improving our understanding of the paper.

Response: We added an extra paragraph explaining more details found on Table 1 (lines 258-260).

  1. I wonder if the authors can improve Figure 5?

Response: We improved figure 5 (lines 343-345): we changed its layout, changed the font size and also standardized the style in accordance with the other figures.

  1. All the remaining parts of the paper are good, I am sure that this is a good paper.

Response: Thank you. We appreciate your comment.

Round 2

Reviewer 3 Report

The authors have improved the paper and it is now ready for publication.